# Clinical Validation of Novel Chip-Based Digital PCR Platform for Fetal Aneuploidies Screening

**DOI:** 10.3390/diagnostics11071131

**Published:** 2021-06-22

**Authors:** Anna Nykel, Rafał Woźniak, Agnieszka Gach

**Affiliations:** 1Department of Genetics, Polish Mother’s Memorial Hospital Research Institute, 93-338 Lodz, Poland; 2Chair of Statistics and Econometrics, Faculty of Economic Sciences, University of Warsaw, 00-241 Warsaw, Poland; rafal.wozniaque@gmail.com

**Keywords:** prenatal diagnosis, copy number variation, aneuploidy screening, chip-based method, digital PCR, sex chromosome abnormality

## Abstract

Fetal aneuploidy is routinely diagnosed by karyotyping. The development of techniques for rapid aneuploidy detection based on the amplification reaction allows cheaper and rapid diagnosis. However, the currently available solutions have limitations. We tested a novel approach as a diagnostic tool in clinical practice. The objective of this study was to provide a clinical performance of the sensitivity and specificity of a novel chip-based digital PCR approach for fetal aneuploidy screening. The study was conducted in 505 pregnant women with increased risk for fetal aneuploidy undergoing invasive prenatal diagnostics. DNA extracted from amniotic fluid or CVS was analyzed for the copy number of chromosomes 13, 18, 21, X, and Y using a new chip-based solution. Performance was assessed by comparing results with findings from karyotyping. Aneuploidy was confirmed in 65/505 cases positive for trisomy 21, 30/505 cases positive for trisomy 18, 14/505 cases positive for trisomy 13 and 21/505 with SCAs. Moreover, 2 cases with triploidy and 2 cases with confirmed mosaicisms of 21 and X chromosomes were detected. Clinical sensitivity and specificity within this study was determined at 100% for T21 (95% CI, 99.26–100%), T18 (95% CI, 99.26–100%), and T13 (95% CI, 99.26–100%). Chip-based digital PCR provides equally high sensitivity and specificity in rapid aneuploidy screening and can be implemented into routine prenatal diagnostics.

## 1. Introduction

Chromosome abnormalities cause birth defects and are present in approximately 3–5% of all clinically diagnosed pregnancies [1]. The most common numerical autosomal abnormalities are trisomy 21 (Down syndrome), trisomy 18 (Edwards syndrome), and trisomy 13 (Patau syndrome) [1]. Birth defects can also be caused by X or Y chromosome abnormality, also known as sex chromosomal aneuploidies (SCAs). The aim of standard prenatal care is the identification of women with increased risk of fetal aneuploidies using biochemical markers and ultrasound markers in the first and second trimester [2,3]. The gold standard for prenatal diagnosis is conventional cytogenetic analyses. Obtaining a karyotyping result based on cell culture of amniotic fluid or chorionic villus samples require approximately two weeks. Thus, a rapid and precise molecular method for aneuploidy detection is desirable. Currently, the most popular rapid diagnostics is based on real-time PCR [4,5,6].

A more advanced approach that allows absolute quantification of nucleic acid with high sensitivity and specificity is digital PCR (dPCR). The method is based on partitioned target DNA molecules in thousands of replicate reactions [7].

After PCR amplification signal from each partition is fluorescently detected and counted as positive or negative, depending on the presence of target DNA molecules. The reactions containing template DNA produce a positive signal which corresponds to the number of DNA template molecules in the sample. Calculation of absolute DNA quantity is based on Poisson distribution [8,9]. Compared to real-time PCR, digital PCR provides absolute quantification of nucleic acids without reference to the standard curve with higher precision, sensitivity, specificity [10,11]. Furthermore, dPCR analysis is independent of the presence of STR alleles in the population. The use of various digital PCR platforms is the focus of scientific research in several fields of medicine, among them obstetrics and gynecology [12].

Previously, in a concept study, we presented a novel approach for detecting fetal aneuploidies using QuantStudio 3D Digital PCR technology [13]. The results of initial research showed that the chip-based method provides statistically significant discrimination between euploid and aneuploid samples, also in low-level mosaic. However, a small study group prevented us from determining the optimal threshold for sample classification and clinical efficacy evaluation. Because no protocols have been established to guide chip-based dPCR in detecting fetal aneuploidies, implementation of a novel method in clinical practice requires assessment of technical performance and identification of key factors influencing the assay. Thus, we have expanded research and validated the method with larger-scale analysis.

This study aimed to evaluate the clinical validation of a novel rapid method for the detection of fetal genetic aberrations in pregnancies using QuantStudio 3D Digital PCR technology. Our approach is based on several years of experience in utilizing this platform for fetal aneuploidies screening.

## 2. Materials and Methods

### 2.1. Sample Collection

The study was conducted from September 2018 to June 2020 at the Polish Mother’s Memorial Hospital Research Institute in 505 pregnant women with increased risk for fetal aneuploidy undergoing invasive prenatal diagnostics on the basis of abnormal biochemical and/or ultrasound results. The study has been approved by the Bioethics Committee of the Polish Mother’s Memorial Hospital Research Institute (74/2018, issue date: 11 September 2018). Amniotic fluid and chorionic villus samples (CVS) collected during amniocentesis and trophoblast biopsy were used in this study. All patients gave written informed consent in accordance with the Declaration of Helsinki.

### 2.2. Workflow of Chip-Based Digital PCR

Digital PCR was performed using QuantStudio 3D Digital PCR System (Life Technologies, Carlsbad, CA, USA). A chip-based workflow is straightforward, and the results are obtained within approximately 4 h. The protocol is composed of six simple steps: DNA isolation, PCR reaction mixture preparation chip loading, amplification in a thermal cycler, florescence detection, and data analysis. The study design is showed in Figure 1. Fetal DNA was isolated automatically from amniotic fluid (4–6 mL) or CVS (1–5 mg) using a MagCore (Cartridge Code 110, elution volume 60 μL), according to the manufacturer’s instructions (RBC Bioscience, Taipei, Taiwan). To quantify and assess the purity of DNA the NanoDrop spectrophotometer (Life Technologies, Carlsbad, CA, USA) was used. Custom labelled probes were designed in a highly conserved region in the constant-copy genome sequences without single-nucleotide polymorphism on chromosomes 13, 18, 21, X, and Y as previously described [14]. For detection of common fetal aneuploidies, three duplex reactions with TaqMan Assays targeting chromosomes 13, 18, 21, X, and Y were performed. Following method validation, the duplex reactions were finally adjusted in pairs: VIC-labelled probe targeting chromosome 13 with FAM-labelled probe targeting chromosome 21, VIC-labelled probe targeting chromosome X with FAM-labelled probe targeting chromosome 18, and with FAM-labelled probe targeting chromosome Y. The proposed molecular solution does not require referring to the number of copies of an additional external reference. Our experience in dPCR optimization shows that combining in one reaction autosomal and sex chromosome targets provides an accurate analysis of chromosome copy numbers, especially in difficult to interpret cases with a copy number change of X and Y chromosomes.

We composed a 15 μL reaction mix using 7.5 μL of 2× QuantStudio 3D dPCR Master Mix, 0.75 μL of 20× TaqMan Assays VIC and FAM labeled-probe and 1–3 μL template sample. Next, samples were partitioned into 20,000 wells on the chip according to the manufacturer’s instructions using QuantStudio 3D dPCR Chip Loader (Life Technologies, Carlsbad, CA, USA). Following chips sealed, PCR was performed in a ProFlex thermal cycler (Life Technologies, Carlsbad, CA, USA) at 96 °C for 10 min, 40 cycles of 60 °C for 2 min, 98 °C for 30 s, and 60 °C for 2 min. The fluorescence signal was measured and analyzed using a QuantStudio 3D dPCR Instrument (Life Technologies, Carlsbad, CA, USA). To exclude contamination, negative template controls (NTCs) was processed in every run for each assay. NTCs included nuclease-free water used in reaction mixes.

### 2.3. Ploidy Status Determination

QuantStudio 3D AnalysisSuite Cloud Software v3.1.6 (Life Technologies, Carlsbad, CA, USA) was used to assign positive and negative signals and convert counts to copies/μL using Poisson statistics. Thresholds were automatically set for each sample. Chip quality and precision measurement for each reaction were performed. The chromosomal ratio for each target chromosome was calculated by dividing the number of copies/μL of the analyzed chromosome by that of the reference chromosome. Each of the three reactions contained assays labelled FAM or VIC. Analysis was based on a comparison of two readings from a thousand wells on the chip.

### 2.4. Statistical Analysis

Calculations were performed in R statistical software v3.6.1 using the FSA library v0.8.30. The chromosomal ratios were compared using a Mann–Whitney U test. The confidence intervals for sensitivity and specificity were calculated by following Brown et al. proposal [15]. Maternal age distribution in the euploid and aneuploid groups was calculated using the Kruskal-Wallis test. To determine a correlation between maternal age, gestational age, and aneuploidies the Dunn test, the Benjamini-Hochberg correction, and Mann–Whitney U test was performed. The threshold for declaring statistical significance was set at *p* < 0.05.

## 3. Results

### 3.1. Study Samples

A group of 505 pregnant women at increased risk of fetal chromosomal aneuploidy undergoing invasive prenatal diagnostics were enrolled for this study. Patient characteristics and indications for invasive testing are summarized in Table 1. The median maternal age was 33.25 (SD, 5.87; range, 16–47) years, and a total of 172 of 505 (34.06%) pregnant women were 35 years of age or older. The median gestational age at the time of prenatal testing was 16.41 (SD, 3.64; range, 11–34) weeks. The most common indications for prenatal testing were ultrasound abnormalities (44.95% of cases) and abnormal maternal serum levels of first trimester markers (36.24%).

In all 505 samples, the chip-based digital PCR screening was performed. The entire process of the study is shown in Figure 2. Analysis was conducted on 382 (75.6%) amniotic fluid samples and on 123 (24.4%) CVS. For all samples, karyotype results were available for final comparison with dPCR results. Samples for which no karyotype results were obtained or were contaminated with maternal cells were excluded from the dPCR analysis.

### 3.2. Clinical Results of dPCR Screening

To validate the accuracy and clinical applicability of the chip-based digital PCR approach, the analysis of 505 clinical samples was evaluated. Of the samples collected in this study, 125 samples were rated abnormal (24.75%) and 380 samples normal (75.25%). 110 cases were classified as the most common autosomal trisomies (T13, T18, T21) which corresponds to 88.70% of all abnormal results, 13 cases tested positive for SCAs (10.40% of abnormal), and in 2 cases male triploidy was diagnosed (1.60% of abnormal). Trisomy 21 was the most commonly diagnosed trisomy identified in 65 cases. The remaining trisomies 18 and 13 occurred at a lower frequency, 30 and 14 cases, respectively. The SCA-positive samples included 12 positive results for monosomy X and 1 for XXXY. Furthermore, in this study, 2 cases with mosaicism were identified. 1 case was classified as monosomy X mosaicism and 1 case as rare trisomy 21 mosaicism. 

In order to analyze the number of copies of chromosomes 13, 18, 21, X, and Y three comparisons were made. Copy number of probes targeting chromosome 13 were compared to probe targeting chromosome 21, X to Y, and chromosome X to 18. This approach provides results for autosomal aneuploidies, but also a comparative analysis of sex chromosome copies compared to the autosomal value. Following method validation with clinical samples, copies from the X chromosome were subsequently compared with copies of the Y chromosome and chromosome 18. This approach improves sensitivity in detecting sex chromosome aneuploidy. In addition, analysis for chromosome 18 requires two intervals and determination of two thresholds due to the presence of one or two copies of chromosome X in euploid samples depending on fetal sex.

Copy numbers of chromosomes 13, 18, 21, X, and Y were compared with the reference chromosome. The chromosomal ratio for the most common aneuploidies were calculated. The chromosomal ratio distribution for the euploid and aneuploid groups differed significantly (*p* < 0.001 for T13, T18, T21, Mann-Whitney U test; Figure 3). The obtained specificity with optimal threshold allowed precise separation and unambiguous discrimination of euploid and aneuploid groups. Ratio values, range, mean values, and standard deviation for each abnormality are shown in Table 2. The confidence intervals for sensitivity and specificity for T13, T18, T21, and X0 were performed (Table 2). All observations were included in calculations of sensitivity, specificity, and their confidence intervals. For T13, T18, T21, and monosomy X the calculations and confidence intervals are consistent. For each abnormality, the method enabled precise separation with 100% sensitivity and specificity (99.26–100% CI for T13, 99.26–100% CI for T18, 99.26–100% CI for T21, 99.26–100% CI for X0). 

### 3.3. Comparison between Clinical Findings Based on Stratification of Demographic Characteristics

We carried out a statistical analysis of the aneuploidy-positive cases based on stratification on demographic characteristics, including the age of pregnant women, gestational age, and indications for prenatal diagnosis. The mean ± SD maternal age of the 125 pregnant women with abnormal and 380 with normal results was 34.41 ± 5.77 and 32.87 ± 5.87 years, respectively. Maternal age in the euploid and aneuploid groups differed significantly (*p* = 0.0061, Mann-Whitney U test). Subsequently, we analyzed maternal age distribution across euploid and aneuploid samples and obtained significant differences (*p* = 0.0012, Kruskal-Wallis test). Finally, we employed a post-hoc test, namely the Dunn test, to determine which frequency of the tested trisomy was correlated with maternal age. The Benjamini-Hochberg correction demonstrated that only in the case of trisomy 21 a strong correlation with maternal age was observed (*p* = 0.0017). The mean ± SD gestational age with abnormal and normal results was 15.32 ± 3.52 and 16.78 ± 3.62 weeks, respectively. There was a significant difference associated with gestational age (*p* = 2.724 × 10^9^, Mann-Whitney U test). 

The most common indications for invasive prenatal diagnosis were positive serum screening and sonographic findings (Table 3). In the group of normal results, both indications occurred with similar frequency (41.57% and 39.73% for serum screening and ultrasound results, respectively). In the group with identified chromosomal abnormalities, the ultrasound abnormality was observed much more frequently than abnormal maternal serum levels of first trimester markers. There was a significant association between the test indication in euploid and aneuploid groups (*p*-value < 0.0001, Fisher’s Exact Test), and indications were unequally distributed in the aneuploid sample (*p*-value < 0.0001, Exact Multinomial Goodness-of-Fit Test).

### 3.4. Comparison between Chip-Based Digital PCR and Karyotyping for Detecting Aneuploidies 

For all 505 cases, validation of chip-based dPCR results was carried out. The concordance rate of dPCR with fetal karyotype was 100% and there were no false positive nor false negative results for the most common fetal aneuploidies. Moreover, two cases of T21 and X0 mosaicism detected with the dPCR method were confirmed by karyotyping. In 9 cases, results qualified as normal in dPCR, yet an abnormal result in the karyotype was obtained. All of the non-detected aberrations were out of the scope of the dPCR method. In 3 cases, balanced translocation was identified, in 2 cases marker chromosome was found, and in 4 cases female triploidy was diagnosed. The comparison between dPCR results and karyotyping for fetal aneuploidies is shown in Table 4. 

## 4. Discussion

This is the first study documenting the clinical effectiveness of the novel approach using QuantStudio 3D Digital PCR platform for the detection of the most common fetal aneuploidies, such as trisomy 21, trisomy 18, trisomy 13, and SCAs. The results obtained from a large study group demonstrate the ability to distinguish the euploid from the aneuploid group with high specificity and precision. Several years of our clinical experience in the assessment of aneuploidy using chip-based dPCR analysis confirms the usefulness of this solution in rapid prenatal diagnosis.

Our study for the first time confirms the clinical effectiveness of the chip-based digital PCR approach for large-scale prenatal screening. We provide promising evidence that a new solution based on digital PCR enables reliable analysis of fetal aneuploidy and can be used as an initial test in clinical practice.

In this study, sensitivity and specificity for the detection of aneuploidy of chromosomes 21, 18, and 13 were all 100% (95% CI, 99.26–100%). With dPCR, the results were obtained for each sample analyzed, indicating high sensitivity regardless of DNA quality. Some discrepancies in the results obtained from the dPCR compared to the karyotype are the consequence of distinct diagnostic profiles for each method. Detection of other chromosomal abnormalities found by a cytogenetic method such as structural chromosomal aberrations and chromosome marker is limited in the molecular dPCR method. In our study, T21 was the most common trisomy with 52% of abnormal results. T18 and T13 were found in 24% and 11.2% of abnormal results, respectively. The results were consistent with the frequency of trisomy reported in the population databases (EUROCAT) [16]. In our cohort, the risk of autosomal aneuploidies increased with maternal age. This is a well-documented phenomenon and was reported in multiple publications [17,18,19,20]. Aneuploidy frequency was higher in CVS than in amniotic fluid samples, most probably due to the correlation of aneuploidy with abnormal ultrasound results, which prompted early first trimester invasive testing.

In clinical practice, quantitative fluorescence PCR (QF-PCR), fluorescent in situ hybridization (FISH), and multiplex ligation-dependent probe amplification (MLPA) are the most frequently used rapid, targeted aneuploidy detection techniques. FISH is an expensive, relatively labor-intensive technique and requires an overnight hybridization step, which delays getting the result [4,6]. PCR-based methods that involve the amplification of polymorphic microsatellite markers or ligated probes are more suited to high throughput testing. MLPA compared to QF-PCR overcomes the potential problem with amplification of polymorphic microsatellite markers, however requires an overnight hybridization step and the quality of prenatal samples affects the results. QF-PCR and MLPA methods require a post-PCR step for quantifying products using an expensive genetic analyzer [6]. The principle of digital PCR is to partition the PCR reaction into thousands of individual reaction wells before amplification and data acquisition at the reaction endpoint. This approach provides several advantages over QF-PCR, including more precise measurements and absolute quantification without the need for a standard curve with higher specificity [10,11]. Moreover, the analysis does not depend on the presence of alleles in the population, which increases the possibility of a wider application of dPCR solution in clinical practice.

The digital nature of the procedure method based on two-dimensional data is automated, which allows obtaining repeatable and highly readable results. Another advantage of the digital PCR approach is the high accuracy despite a small volume of the initial sample, which is particularly important in difficult to acquire prenatal samples. Digital PCR workflow is straightforward, requires one pre-PCR pipetting step, and the risk of potential contamination is reduced by closing the reaction mixture in a sealed chip. Moreover, the analysis does not require additional bioinformatics tools, because the chip quality and absolute quantification based on Poission distribution are available in the cloud-based software. Several types of digital PCR platforms are available based on the microfluidic droplet technique and chip-based technique [11,12,21,22]. In comparison to droplet dPCR, the QuantStudio dPCR platform provides a constant number of independent reactions of thousands of wells and the results are not dependent on the efficiency of the generated droplets. Although the costs of reagents for chip-based dPCR are not the lowest compared to other techniques used ($10 per sample versus $2 in QF-PCR and $5 in droplet PCR), the equipment costs are two times lower in comparison to other available platforms ($45 K) [21], which ensures easy implementation of the technology in small and medium-throughput laboratories. Moreover, our solution based on the analysis of five chromosomes limited to three duplex reactions reduces the reagent costs for the analysis of the most common aneuploidies. Although dPCR platforms are currently more expensive than QF-PCR for a test with specialized instrumentation and consumables, it provides a more accurate and sensitive analysis, which is especially important in prenatal diagnosis.

Compared to our previous work [13]. which was primarily based on method validation, the current study focused on optimization and evaluation of its clinical effectiveness. Experience in the assessment of the number of chromosome copies allowed to optimize the compared pairs of targets on the chromosome in one reaction. In addition, a new approach based on single-target fetal aneuploidy analysis without an additional probe targeting the reference chromosome required confirmation in a larger study group. The results of our research and several years of clinical practice confirm the usefulness of the single-target analysis per chromosome and indicate that the dPCR method is effective and precise even for samples with mosaicism. A significant advancement over prior work [13]. allowed to determine the exact cut-off point for the identification of aneuploidy in clinical practice and verifies the usefulness of using this analysis on a large scale.

A significant limitation of the current study was chromosomal analysis on a research group from only one center on the Polish population. Despite the large number of prenatal samples used in this study, a relatively low incidence of trisomy 13 and mosaicism was observed. Thus, extensive validation on a much larger group is needed to accurately determine the detection rate of rare aberrations and to assess efficacy in different populations. However, it appears that as the method of analysis is based on known copy-number sequences, as opposed to the polymorphism-based approach, the results of this study are representative. Another limitation of dPCR analysis, similarly to aCGH, is the inability to identify female triploidy. In contrast, the method identifies male triploidy due to differences in sex chromosome copies. In addition, dPCR is limited by low throughput enabling analysis of 8 patients within 4 h. However, the rapid test is carried out on an ongoing basis and this capacity should be sufficient for small or medium size laboratories.

In summary, our study demonstrates that QuantStudio 3D Digital PCR is a reliable and accurate prenatal technique to identify fetal chromosomal abnormalities. A single-molecule-amplification with high sensitivity and specificity, minimum hands-on time methodology, and potential to determine clinically significant abnormalities with high precision makes the dPCR approach reliable and cost-effective rapid initial test. Although the rapid test is a preliminary step while awaiting the result of full karyotype, the high incidence of aneuploidies in the study cohort underlines the effectiveness, reproducibility, and importance of dPCR analysis prior to additional testing. Hopefully, the results from this study will contribute to improvements in prenatal diagnostics and open up opportunities for further research on the perspectives and limitations of chip-based dPCR as a diagnostic tool.

## Figures and Tables

**Figure 1 diagnostics-11-01131-f001:**
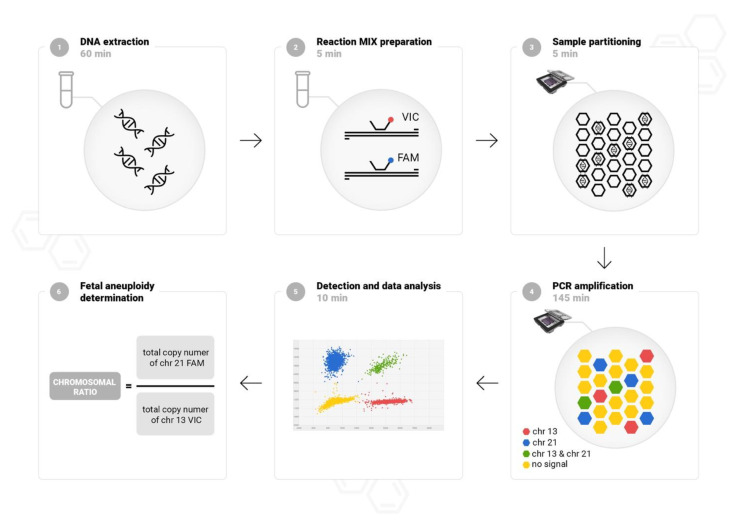
The procedural framework of chip-based digital PCR method for fetal aneuploidies detection. The procedure time is approximately 4 h. The process is based on the 6 following steps: fetal DNA from amniotic fluid samples or CVS extraction (1), dPCR reagents preparation (2), sample partitioning (3), PCR amplification (4), fluorescence detection (5) and aneuploidy status determination (6). The hexagons represent the thousands of reaction wells on the chip. The example of duplex reaction presented consists of assay targeting chromosome 13 labeled with VIC dye and assay targeting chromosome 21 labeled with FAM dye. The scatter plot represents signals from FAM and VIC dye from wells on the chip. Four colors in the plot corresponds to the FAM call (blue), VIC call (red), both FAM and VIC calls (green) and no signal (yellow).

**Figure 2 diagnostics-11-01131-f002:**
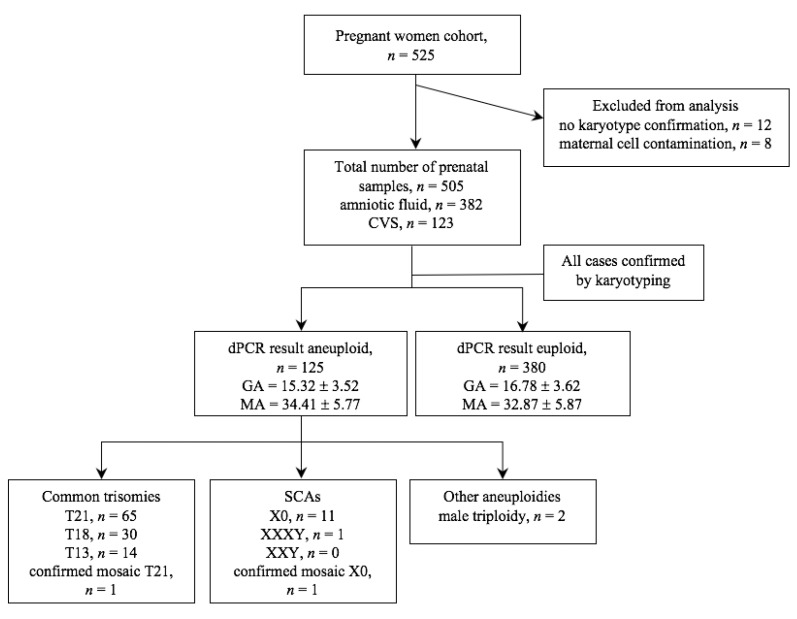
Flowchart of chip-based digital PCR aneuploidies screening. Figure showing study cohort, excluded samples and test results. Values are given as *n* or mean value ± standard deviation. T13, trisomy 13; T18, trisomy 18; T21, trisomy 21; SCAs, sex-chromosomal abnormalities; CVS, chorionic villus samples; GA, gestational age; MA, maternal age.

**Figure 3 diagnostics-11-01131-f003:**
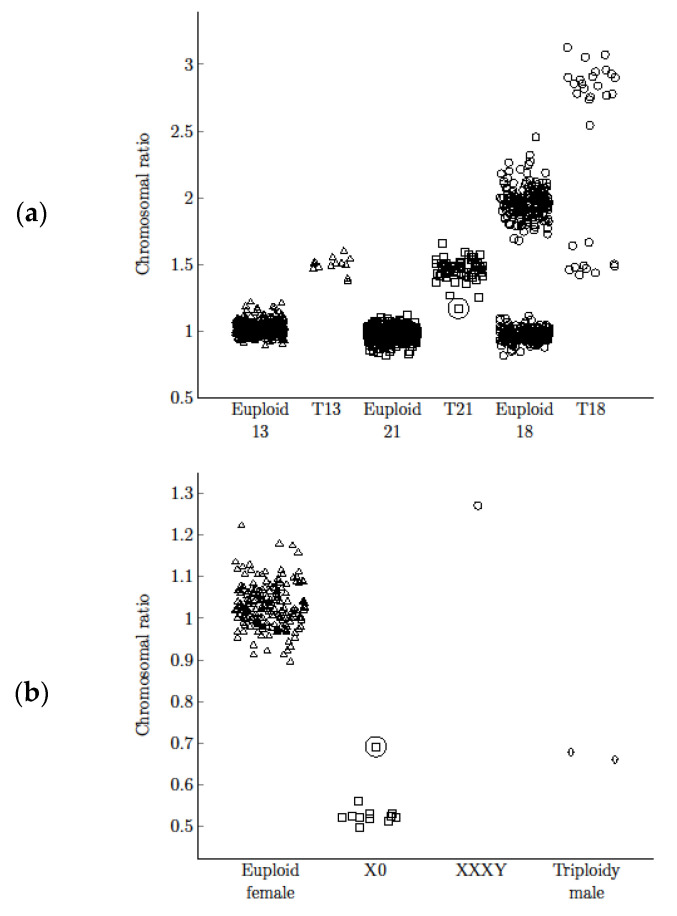
Chromosomal ratios distribution in euploid and aneuploid group. Upper panel represents chromosomal ratio in most common aneuploidies (T13, T21, T18 female on the upper dataset and T18 male on the lower dataset) (**a**). Lower panel represents chromosomal ratio distribution in samples with abnormal X chromosome copy number compared to the ratio value for euploid female samples (**b**). Two samples marked in the circle correspond to cases with identified T21 (**a**) and X0 mosaics (**b**).

**Table 1 diagnostics-11-01131-t001:** Demographic and clinical characteristics of 505 pregnant women undergoing invasive prenatal testing.

Characteristic	Total Population n (%)
Maternal age	33.25 ± 5.87 (16–47)
<35 years	333 (65.94)
≥35 years	172 (34.06)
Gestational age	16.41 ± 3.64 (11–34)
First trimester (9–13 weeks)	71 (14.06)
Second trimester (14–27 weeks)	379 (75.05)
Third trimester (28–40 weeks)	15 (2.97)
Singleton pregnancy	495 (98.02)
Twin pregnancy	10 (1.98)
Clinical features	
Positive serum screening	183 (36.24)
Ultrasound abnormality	227 (44.95)
Two or more indications	59 (11.68)
Family history	8 (1.58)
Other or not indicated indications	28 (5.54)
Diagnostic procedure	
CVS	123 (24.4)
Amniocentesis	382 (75.6)

Values are given as *n* (%) or mean value ± standard deviation. Gestational age data were not available for every pregnancy, therefore do not sum to 100%.

**Table 2 diagnostics-11-01131-t002:** Screening performance for indicated abnormalities in the current cohort.

Abnormality	Ratio for Aneuploid Samples	Ratio for Euploid Samples	Sensitivity (95% CI)	Specificity (95% CI)
T13	1.495 ± 0.058 (1.369–1.600)	1.018 ± 0.049 (0.889–1.217)	100% (99.26–100%)	100% (99.26–100%)
T18 female	1.507 ± 0.081 (1.423–1.666)	0.975 ± 0.048 (0.818–1.117)	100% (99.26–100%)	100% (99.26–100%)
T18 male	2.870 ± 0.131 (2.544–3.126)	1.965 ± 0.113 (1.679–2.454)	100% (99.26–100%)	100% (99.26–100%)
T21	1.468 ± 0.067 (1.258–1.659)	0.984 ± 0.046 (0.821–1.125)	100% (99.26–100%)	100% (99.26–100%)
X0	0.522 ± 0.015 (0.497–0.560)	0.975 ± 0.048 (0.818–1.117)	100% (99.26–100%)	100% (99.26–100%)

Values are given as mean value ± standard deviation (range). CI, confidence interval; T13, trisomy 13; T18, trisomy 18; T21, trisomy 21; X0, monosomy X.

**Table 3 diagnostics-11-01131-t003:** Comparison between indications and ploidy status.

Sample Group	N	Indications				
		Serum Screening	Ultrasound Abnormality	Family History	Two or More Indications	Not Specified
Euploid	380	158 (41.57)	151 (39.73)	8 (2.10)	38 (10.0)	26 (6.82)
Aneuploid	125	25 (20.0)	76 (60.80)	0 (0)	21 (16.80)	2 (1.60)
T13	14	3 (21.42)	10 (71.42)	0 (0)	1 (7.14)	0 (0)
T18	30	4 (13.33)	18 (60.0)	0 (0)	8 (26.66)	0 (0)
T21	65	17 (26.15)	34 (52.30)	0 (0)	12 (18.46)	2 (3.07)
X0	12	0 (0)	12 (100)	0 (0)	0 (0)	0 (0)
Others	3	1 (33.33)	2 (66.66)	0 (0)	0 (0)	0 (0)

Values are given as *n* (%).

**Table 4 diagnostics-11-01131-t004:** Comparison between chip-based digital PCR and karyotyping results.

Sample ID	Age	Gestational Weeks	Ultrasound Indication	Serum Screening	Cytogenetic Result	dPCR Result
Case 1	19	17 W	abnormal	low risk	69, XXX	euploid female
Case 2	36	16 W	abnormal	low risk	46, XY, t(6;8)(q23;q21)	euploid male
Case 3	39	12 W	abnormal	high risk	69, XXX	euploid female
Case 4	28	15 W	normal	high risk	46, XY, +21, der(21;21)(q10;q10)	T21 male
Case 5	40	17 W	normal	high risk	46, XX, t(9;17)(p10;p10)	euploid female
Case 6	32	14 W	abnormal	low risk	69, XXX	euploid female
Case 7	21	21 W	abnormal	low risk	46, XY, +21, der(14;21)(q10;q10)	T21 male
Case 8	27	14 W	abnormal	low risk	69, XXX	euploid female
Case 9	30	15 W	normal	high risk	mos 46, XX, t(7;8)(q11.2;q24.1) [5]/46, XX [6]	euploid female

## Data Availability

The datasets presented in this study are available from the corresponding author. The data are not publicly available due to the individual private information.

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
