# Peer review of "Clinical Validation of Novel Chip-Based Digital PCR Platform for Fetal Aneuploidies Screening"

_diagnostics, 2021, doi:10.3390/diagnostics11071131_

Round 1
Reviewer 1 Report
The authors report on the application of a previously developed method in a relevant number of cases for the detection of aneuploidy for chromosomes 13, 18, 21, X and Y use digital PCR.
The results are convincing and the experimental environment is solid and I have no major criticisms.
In view of the aim of the work stated by the authors, i.d. the applicability of this method as a first line test for the rapid detection of aneuploidy, however, the authors did not really thoroughly work out the usefulness of this new method in comparison to existing alternatives such as FISH, QF-PCR and microsatellite analysis. Rapid detection of aneuploidy can be achieved using a set of microsatellites. This method is already implemented in practically every laboratory that performs prenatal karyotyping as a maternal contamination test and in some laboratories as a rapid test for the most common aneuploidies. In some cases, microsatellites may not be informative (i.e., in the case of parental consanguinity), although the allele ratio may be indicative of the presence of a trisomy.
The clinical advantages and costs of digital PCR compared with established methods should be considered in more detail in the discussion, as well as the practicality of introducing this method for small, medium-sized laboratories, which represent the vast majority of laboratories in many countries. The authors indicated that the results of their study could or may help improve the current diagnostic prenatal routine without clarifying what kind of improvements will be possible with this method. I was wondering what bioinformatics / statistical skills are needed for the calculation and to what extent knowledge of the R programming language or the Linux operating system is truly needed for smoothly working. These aspects should be taken into account in seriously considering a practical implementation of digital PCR in a routine environment. I would suggest that the authors give to the readers a realistic picture of the cost in terms of required equipment and personal skills compared to the methodology currently in use.
Author Response
Specific Comments to Reviewer 1
We thank the Reviewer for the useful suggestions, which have allowed us to clarify the manuscript.
We are very grateful to the Reviewer for favourable opinion. We appreciate all suggestions and we added a paragraph to the discussion comparing digital PCR to the most commonly used methods: FISH, QF-PCR, MLPA (line 297-334). In addition, we have included pricing and commentary on reagent and equipment costs for further comparison of QuantStudio 3D dPCR and other diagnostic solutions for fast aneuploidy analysis (line 326-334)
Reviewer 2 Report
It is a very clear manuscript about a novel screening test for trisomy 21, 13, 18, and X and Y chromosomal abnormalities- chip-based digital PCR approach. This screening test provides high sensitivity and specificity , with a result available in 4 hours , and can be implemented into routine prenatal diagnostics.
I have three questions for the authors :
- What is the value of this test for twins- monozygotic and dizygotic
- Is the fetal fraction important for validating results?
- You say it is a “cheap test “ – what is the cost for one test and the optimal implementation?
For lines 37,75 : CVS means chorionic villus sample instead of chronic
Author Response
Specific Comments to Reviewer 2
We thank the reviewer for comments and questions, which have allowed us to further improve this work.
- What is the value of this test for twins- monozygotic and dizygotic
We thank the reviewer for the question. In our clinical practice we identified 10 dizygotic twin pregnancies (Table 1). In the case of samples from a monozygotic twin pregnancy, the analysis should be similar to that for the presence of fetal mosaicism. Our method has a detection limit of less than 20%.
- Is the fetal fraction important for validating results?
We thank the reviewer for the question. Determination of the fetal fraction was important in the validation of the method, especially in the case of samples with mosaicism and contamination with maternal material. In our practice, we have correctly identified 2 cases of mosaicism. Based on the Y chromosome analysis for each sample, the calculated ratio indicated a possible value of the fetal fraction lower than 100%.
- You say it is a “cheap test “ – what is the cost for one test and the optimal implementation?
We thank the reviewer for the suggestion. We added some details for instrument and sample cost (line 326-334 )
- For lines 37,75: CVS means chorionic villus sample instead of chronic
We changed “chronic” to “chorionic” (lines 37, 75, 175)